# An Automatic Detection and Classification System of Five Stages for Hypertensive Retinopathy Using Semantic and Instance Segmentation in DenseNet Architecture

**DOI:** 10.3390/s21206936

**Published:** 2021-10-19

**Authors:** Qaisar Abbas, Imran Qureshi, Mostafa E. A. Ibrahim

**Affiliations:** 1College of Computer and Information Sciences, Imam Mohammad Ibn Saud Islamic University (IMSIU), Riyadh 11432, Saudi Arabia; meibrahim@imamu.edu.sa; 2Department of Computer Software Engineering, Military College of Signals, National University of Sciences and Technology (MCS-NUST), Islamabad 44000, Pakistan; imarwat11@gmail.com; 3Department of Electrical Engineering, Benha Faculty of Engineering, Benha University, Qalubia, Benha 13518, Egypt

**Keywords:** retinal fundus images, diabetic retinopathy, hypertensive retinopathy, deep-neural network, semantic and instance-based segmentation, transfer learning, perceptual-oriented color space, DenseNet architecture, loss function

## Abstract

The stage and duration of hypertension are connected to the occurrence of Hypertensive Retinopathy (HR) of eye disease. Currently, a few computerized systems have been developed to recognize HR by using only two stages. It is difficult to define specialized features to recognize five grades of HR. In addition, deep features have been used in the past, but the classification accuracy is not up-to-the-mark. In this research, a new hypertensive retinopathy (HYPER-RETINO) framework is developed to grade the HR based on five grades. The HYPER-RETINO system is implemented based on pre-trained HR-related lesions. To develop this HYPER-RETINO system, several steps are implemented such as a preprocessing, the detection of HR-related lesions by semantic and instance-based segmentation and a DenseNet architecture to classify the stages of HR. Overall, the HYPER-RETINO system determined the local regions within input retinal fundus images to recognize five grades of HR. On average, a 10-fold cross-validation test obtained sensitivity (SE) of 90.5%, specificity (SP) of 91.5%, accuracy (ACC) of 92.6%, precision (PR) of 91.7%, Matthews correlation coefficient (MCC) of 61%, F1-score of 92% and area-under-the-curve (AUC) of 0.915 on 1400 HR images. Thus, the applicability of the HYPER-RETINO method to reliably diagnose stages of HR is verified by experimental findings.

## 1. Introduction

The most common cause of retinal damage is hypertensive retinopathy (HR). By 2025, 1.56 billion people are projected to suffer from hypertension. Besides that, nearly 66% of people who are affected by hypertension reside in developing or impoverished countries, where the lack of sufficient healthcare services to identify, manage and handle hypertension exacerbates the issue [1]. Hypertension triggers nosebleeds, vision loss and headaches. Moreover, long-term hypertension can cause permanent damage to the lungs, heart, kidneys and eyes. Among all these consequences, hypertensive retinopathy (HR) is perhaps the most common cause of cardiovascular disease, which results in death. As a result, it is recognized as a worldwide community health hazard. The risk of HR can be reduced if hypertension is identified and treated early. It is difficult to diagnose hypertensive retinopathy in the early stages due to the lack of advanced imaging technology and proficient ophthalmologists [2].

Hypertensive retinopathy (HR) is a retinal deformity caused by elevated blood pressure in general. The appearance of arteriolar narrowing [3], arteriovenous nicking, retinal hemorrhage (HE), microaneurysms, Cotton wool spots (CWS), papilledema and, in severe cases, optic disc and macular edema are other significant symptoms of HR-related eye disease. Mild hypertensive retinopathy symptoms are general, according to reports, and are seen in approximately 10% of the non-diabetic adult populace [4]. Recently, numerous studies have documented that the microscopic fundus images taken by an optical camera can be used to visualize retinal microvascular disorders. Numerous HR patients are safely inspected with this fundus camera since it is cheap, easy to perform and shows most clinical lesion structures in its fundus images [5]. Categorizing HR into different grades is a difficult task for computerized diagnostic systems, and, to the best of our knowledge, no research has assessed HR stages (Grade 0, Grade 1, Grade 2, Grade 3 and Grade 4) corresponding to normal (no sign of abnormalities), mild, moderate, severe and malignant. Figure 1 gives visual examples of the five different stages of HR.

To classify HR, there are two kinds of computerized systems. One category is associated with methods focused on complex low image processing. The second kind concerns deep-learning (DL) models, which are used to automatically extract features along with preprocessing and image processing algorithms. DL-based approaches achieved significantly higher accuracy as compared to features derived using complex image processing systems. However, to prevent over-fitting with a small dataset of HR, certain DL networks must be fine-tuned. Researchers have introduced a few state-of-the-art automated detection systems for detecting HR-related diseases based on two stages in the literature. These systems are briefly listed and compared in Section 2. Although few approaches have been established for classifying retinal fundus images into two-category-based HR and non-HR, none of them focused on five categories of HR. To define HR characteristics, it is arduous to specify and recognize relevant HR lesion properties from fundus images. There are no datasets available, and we need medical specialists (ophthalmologists) to identify the HR stages directly from retina fundus images and to train the network. Some DL variant models are designed to automatically learn characteristics, but at each layer, they utilize the same-weight scheme. For accurate decisions, it is hard for layers to transfer weights to deeper network levels. There is a need to develop an automated solution for recognizing the five stages of HR, and to the best of our knowledge, no study has addressed the five severity-level of HR.

### Research Highlights

Few approaches are established for classifying retinal fundus images into two-category-based HR and non-HR, none of them focused on five categories of HR. To define HR characteristics, it is arduous to specify and recognize relevant HR lesion properties from fundus images. Therefore, in this paper, we have developed an automatic detection and classification system (HYPER-RETINO) of five stages for hypertensive retinopathy using semantic and instance segmentation in DenseNet architecture. The main contributions to this paper are given as follows.

A preprocessing step is integrated to build this HYPER-RETINO system to adjust light illumination and enhance the contrast in a perceptual-oriented color space.A novel semantic and instance-based segmentation mechanism is introduced to classify and identify the HR-related lesion’s pixels and regions.This deep-learning network has been trained with a wide set of HR fundus images to improve the HYPER-RETINO system to ensure the better applicability of this model.

## 2. Literature Review

To identify retinal irregularities, many automated systems have previously been proposed. In contrast, there are only a few automated systems to detect HR retinopathy and even fewer systems to classify the HR retinopathy into severity levels. Recently, several researchers have used a retinal fundus image processing technique to diagnose the HR disease automatically. The early identification of HR disease by fundus image processing saves ophthalmologists a lot of work and time [6,7,8,9]. State-of-the-art research in this field is presented in this section. The state-of-the-art papers are divided into three categories: anatomical structure-based approaches, traditional machine and deep learning methods.

### 2.1. Anatomical Structures-Based Techniques

To classify hypertensive retinopathy (HR) from microscopic retinography images, the literature review suggests the usage of segmentation-based methods [6]. In such techniques, first, various HR-related characteristics are identified and then used as inputs for a traditional machine learning classifier to identify the HR retinal fundus among images of the color fundus.

The handcrafted features used in automated systems to identify retinal anomalies such as HR different grades and the bifurcation of retinal blood vessels in [7,8,9,10,11,12,13] are the arteries and veins diameter ratio (AVR), optic disc (OD) position, mean fractal dimension (mean-D), papilledema signs and tortuosity index (TI). For segmentation and low-level operations, the Gabor 2D or cake wavelet and the canny edge detection scheme are used. To verify the effectiveness of such systems, INSPIR-AVR, arteriolar-to-venular diameter ratio database (AVRDB), VARPA images for the computation of the arterio/venular ratio (VICAVR), structured analysis of the retina (STARE), digital retinal images for vessel extraction (DRIVE), diabetic retinopathy hypertension age-related macular degeneration and glaucoma images database (DR-HAGIS) and IOSTAR datasets have been used. In many studies, preliminary segmentation and refining phases are conducted with a supervised classifier to classify clusters of hemorrhages. A specific method for identifying HR-related retinal disorder is used in [14]. Cotton wool spots (CWS) are identified and proved to be a significant clinical symptom for identifying HR-related retinal diseases. To improve the suspected areas, it uses the Gabor filter bank followed by the thresholding technique to convert the image into binary. This method achieves a sensitivity of 82.2% and a PPV of 82.38% using local fundus images. Five forms of retina abnormalities, namely, Diabetic Retinopathy (DR), Hypertensive Retinopathy (HR), Vitreous Hemorrhage (VH), Venous Branch Occlusion (VBO), Macular Degeneration (MD) and Normal Retina, are recognized in the complete system [15]. They use a wavelet-based neural network technique with initially handled images to detect all of the retina abnormalities. For the performance assessment of this method, five datasets are utilized achieving minimum and maximum accuracies of 50% and 95% according to the quality of the fundus images of those datasets.

In [16,17,18,19], the arteriolar-to-venular diameter ratio (AVR) technique is implemented on chosen tiny groups of color fundus images from massive datasets including diabetes control and complications trial (DCCT) and early treatment diabetic retinopathy study (ETDRS). The AVR ratio is determined either by the measurement of vessel diameters by Gabor wavelet, gradients and operation of morphological edge detection or by the separation of the OD area and the differentiation between arteries and veins. In [20], a graphical interface for the semi-automated detection and measurement of retinal vessels to classify HR-related retinal disorders has been created. To aid vascular risk in individuals with HR, this graphical user interface (GUI) framework can be used to quantify vessel widths at any region of interest (ROI). Digital retinal images for vessel extraction (DRIVE) and structured analysis of the retina (STARE) datasets are employed for the performance evaluation of the system achieving comparable results. A clustering technique along with AVR is utilized in [21] for vessels segmentation and classification. In [22], gray-level and moment features are derived to identify pixels contributing to the blood vessel, and intensity variance and color details are used to distinguish vessels into arteries or veins. By estimating vessel diameter on 101 images obtained from the VARPA images for the computation of the arterio/venular ratio (VICAVR) dataset, various stages of HR-related retinal deficiencies are categorized. From the VICAVR dataset, 76 images are used as HR candidates, while 25 images are identified as normal candidates. To separate blood vessels in the retina, the Hough and Radon transformations have been utilized in [23]. Next, the authors measure the TI and the vessel’s width. With the aid of these measurements, the AVR is computed and used for identifying HR in microscopic retinography images. The researchers in [24] employ independent component analysis (ICA) on wavelet sub-bands to discover the various abnormalities that exist in the fundus of the retina, such as optic discs, blood vessels, hemorrhage clusters, macula and exudates. Fifty retinal fundus images are employed for assessing the performance of this approach with a noticeable accuracy. In [25], the authors calculate the invariant moments along with Gabor wavelets from DRIVE retinal fundus images. Then, they use them as inputs to the neural network (NN) to categorize blood vessels into arteries and veins. Their results show superior classification accuracy. An automated multi-stages method is built-in [26] to locate the OD area, segment vessels, detect color characteristics, estimate the AVR ratio, distinguish vessels into veins or arteries, compute the average red intensity and then differentiate the candidate retinal fundus images into HR or normal. An area under the receiver operating curve (AUC), specificity (SP), and sensitivity (SE) are used as performance metrics. These metrics are computed for up to 74 images by showing AUC of 0.84, SE of 90% and SP of 67%. In [27], a software package called quantitative analysis of retinal vessel topology and size (QUARTZ) has been developed to segment vessels, measure their thickness and tortuosity index, identify OD location and finally categorize vessels into arteries or veins. The DRIVE dataset is used for assessing its performance and it achieves an accuracy of 84%.

### 2.2. Traditional-Machine and Deep-Learning Based Techniques

Few HR systems have also been developed in the past to classify two stages by using traditional-machine and deep-learning-based techniques. Those HR systems are briefly described in the subsequent paragraphs.

HR is identified in [28] by feature extraction from initially processed color retina fundus images. The first initial fundus image processing step is to transform them using contrast limited adaptive histogram equalization (CLAHE) to the green channel, which clarifies the vessels’ view. Next, the optic disc (OD) is separated using morphological closing. Then, by applying subtraction, the fundus background is removed, and features are identified by employing zoning. Lastly, the extracted features are used to train a feed-forward artificial neural network classifier. The classifier results in an accuracy of 95%. In [29], the authors used an extreme learning machine (ELM) classifier for segmenting retinal fundus vessels. It is trained with the aid of 39 features. A segmentation accuracy of 96.07%, specificity of 98.68% and sensitivity of 71.4% are achieved using the DRIVE dataset. Recently, deep learning (DL) approaches have been frequently utilized for retinal fundus images classification. The DL methodology is characterized by minimal input image pre-processing requirements. Numerous image processing stages, such as the extraction of low-level features and segmentation, are implied in the DL architecture. The authors of [30] introduce an early trial for the recognition of HR using a convolutional neural network (CNN) as a DL model. The CNN is fed with a stream of (32 × 32) bunches of the grey level transformed version of the original retina fundus images. Their approach identifies the input fundus images into either HR or normal with an identification accuracy of 98.6%. Another recent hypertensive retinopathy (Dense-Hyper) identification approach is implemented in [31]. Their approach employs a trained features layer and a dense feature transform layer into the deep residual learning architecture. It is used to categorize two classes of HR. Therefore, it is limited in its ability to apply for assistance to ophthalmologists for recognition of five stages of HR.

Many researchers employ the DL architectures in mid-level tasks, such as retina vessels segmentation or optic disk separation, that are necessary for high-level tasks such as the classifications of DR or HR. A DL architecture [32] composed of a deep neural network (DNN), and a random Boltzmann machine (RBM) is used for quantifying any alteration in retinal blood arteries vessels using AVR ratio and OD region determination. They achieve a good detection accuracy. In [33], the authors present an approach to locate the fovea center relative to the OD position using 7-layer CNN architecture. Their system output consists of a four nodes layer that signifies the fovea center, OD, retina blood vessel and retina background. The system uses the DRIVE dataset for performance assessment and results in a classification accuracy of 92.68%. Other trials [34,35,36] use CNN for the segmentation of retinal veins and arteries. Those trials employ a low-quality dataset of 100 fundus images and the DRIVE datasets for performance evaluation, and they achieve accuracies of 88.89% and 93.5%, respectively. An automated CNN-based approach for the detection of exudates in the microscopic retinal fundus images is presented in [37]. Throughout the CNN training process, the features are comprehensively extracted. The CNN inputs are odd-size bunches where the pixel in the middle of the bunch is the one under processing. Convolution layers are used to identify the likelihood of exudation or not exudation of each pixel. Since no exudates appear around the optic disk, the OD area is removed. In addition to the input and output layers, the CNN model consists of four convolving and pooling layers. The system is tested using the diabetic retinopathy image dataset (DRiDB) dataset and shows a 77% F-Score.

Recently, the authors in [38] developed the Arsalan-HR system to detect vessels from retinograph images using a dual-residual-stream-based method. The authors used semantic segmentation to recognize HR and non-HR stages of hypertensive retinopathy in a deep-learning architecture with few parameters. The authors tested the Arsalan-HR system on three publicly available datasets such as DRIVE, the child heart health study in England (CHASE-DB1) and STARE on limited datasets without considering the multistage of HR. In addition, they did not consider the pre-processing step to enhance the contrast of vessels. However, many HR-related lesions are important to detect for the recognition of the stage of HR. Similarly, in [39], the authors developed a Kriplani-AVR system to recognize two stages of HR based on the AVR ratio of blood vessels. The Kriplani-AVR system was tested on the DRIVE dataset. In that study, the authors aggregated the residual neural network with CNN and achieved 94% detection accuracy. More recently, in [40], the authors developed a Tang-Semantic system based on semantic segmentation by a CNN architecture for the detection and localization of diabetic retinopathy (DR)-related lesions. The Tang-Semantic approach is useful to detect DR, but it cannot be applied to detect HR due to the AVR ratio. The authors detected only DR-related lesions without an AVR ratio. As a result, it is limited in its capacity to detect all stages of HR.

## 3. Materials and Methods

The major three stages are integrated (as shown in Figure 2) to develop this HYPER-RETINO grading system for hypertensive retinopathy (HR). These are preprocessing steps to enhance the HR-related lesions, the detection of lesions by semantic and instance-based segmentation and grading through a dense feature transform layer by the four dense blocks. Finally, a classification decision is performed based on the SoftMax layer to predict five stages of HR. Overall, the HYPER-RETINO system is used to determine the local regions within input retinal fundus images to recognize various stages of HR. This section demonstrates the proposed framework for recognizing and classifying of five-stage of HR.

### 3.1. Acquistions of Datasets

A recognition system of grading for hypertensive retinopathy (HYPER-RETINO) is evaluated in terms of performance and compared with state-of-the-art HR systems in terms of five stages of hypertensive retinopathy by collecting different retinograph image datasets. It is pertinent to mention that there is no study available that uses an online dataset for the recognition of multi-stage HR. As a result, we requested two experienced ophthalmologists to create a gold standard for the identification of five stages of HR using private and public data sources. In total, 1400 images are collected from one private (PRV-HR) and six online data sources (DR-HAGIS [12], DRIVE [41], DiaRetDB0&1 [42], DR1&DR2 [43], Kaggle-DR [44] and APTOS-DR [45]), as described. In this five stage dataset, the division of each class is as follows: normal of 400 images, mild-HR of 200, moderate-HR of 200, severe-HR of 200 and malignant-HR of 400 retinograph images. These retinograph images are of various sizes. Therefore, to make standard size, we have resized all retinograph images to (700 × 600) pixels in JPEG format. A visual example of this dataset is displayed in Figure 1. If the data augmentation technique is applied, then we can be able to increase the size of the acquired dataset. However, we applied the data augmentation technique offline in this paper to increase the size of the dataset. In the data augmentation step, the flipping (horizontally + vertically) and rotation from 90 degrees to 180 degrees without scaling are applied. This increases the number of training images from 1400 to 2800. The retinograph images in these datasets have different light illumination and unclear HR-related lesions. Therefore, we have performed a pre-processing step developed in this paper to enhance the HR-related lesions with light and contrast adjustment in a perceptual-oriented color space. After the pre-processing step, the HR-related lesions are identified by two expert ophthalmologists for providing us with ground truth. To generate a ground truth mask, we used a free online annotation tool. These ground truth masks help to compare the performance of proposed semantic and instance-based segmentation techniques. In this paper, the size of the mask is fixed by resizing the ground truth of retinograph images to (700 × 600) pixels. To pretrain the DenseNet network, these HR-related regions are extracted from each retinograph image.

### 3.2. Proposed Methodology

To develop this HYPER-RETINO system, we have developed several steps, which are explained in the subsequent subsections. It was noticed that the retinograph images in these datasets had different light illumination and unclear HR-related lesions. Therefore, we have performed a preprocessing step developed in this paper to enhance the HR-related lesions with light and contrast adjustment in a perceptual-oriented color space. After the pre-processing step, the HR-related lesions are detected from each image through a combination of semantic and instance segmentation techniques, and then these HR-related regions are classified by a pretrain DenseNet architecture. Those steps are explained in the subsequent subsections and visually represented in Figure 2.

#### 3.2.1. Preprocessing in Perceptual-Oriented Color Space

The purpose of the preprocessing phase is to reduce the impact of lighting differences between images, such as the brightness and angle of incidence of the fundus camera. This preprocessing is carried on in two steps such as color space transformation, the correction of the lightening and the enhancement of the contrast. By converting retinal-colored fundus images into gray-scale images, much useful information has vanished. As a result, the classification of spatial information presented on pixels is lost. Therefore, it is important to represent retinograph images in a perceptual-oriented color space, which considers the viewing conditions. In this paper, we have selected the CIECAM02 color appearance model [46] because it is more advanced compared to other color spaces. In practice, the CIECAM02 provided the most advanced features, including six dimensions of color appearance: brightness (Q), lightness (J), colorfulness (M), chroma (C), saturation (s) and hue (h). This color appearance model is unable to construct a true-color space, so the original input is transformed to CIECAM02 space with lightness, chroma and hue correlates (J, C, h). Hence, the JCh color planes are utilized to enhance the image. Compared to other color spaces such as HSV, CIELUV or CIELab, the JCh color space is not completely uniform, but it provided most of the uniformity to account for all of the perceptual phenomena. To improve the uniformity, it has advanced metrics very similar to CIEDE2000 in CIELab.

Accordingly, the JCh uniform color space is the color system adopted by the proposed method because of its better uniformity and adaptation to human perception. Thus, the luminance improvement (J-plane) process must be carried on correctly to confirm that the improved images retain the correct color information. This can be accomplished by obtaining a luminance gain matrix LG (*α*, *β*) on a J-color plane image as follows:(1)$$G\left(\alpha ,\beta \right)=\frac{r^{\prime }\left(\alpha ,\beta \right)}{r\left(\alpha ,\beta \right)}=\frac{g^{\prime }\left(\alpha ,\beta \right)}{g\left(\alpha ,\beta \right)}=\frac{b^{\prime }\left(\alpha ,\beta \right)}{b\left(\alpha ,\beta \right)}$$
where *r*′, *g*′ and *b*′(*α*, *β*) are the values of the RGB components of a pixel in the improved fundus image at (*α*,*β*) coordinates, while *r*, *g* and *b*(*α*,*β*) are the RGB values of original fundus image at (*α*,*β*) coordinates. The luminance matrix is defined as shown in Equation (2) for the color invariant improvement of an RGB image.
(2)$$LG\left(\alpha ,\beta \right)=\frac{\partial^{\prime }\left(\alpha ,\beta \right)}{\partial \left(\alpha ,\beta \right)}=\frac{\partial^{\prime }\left(\alpha ,\beta \right)}{\sqrt{\sum_{c\in \left\{r,g,b\right\}}c^{2}\left(\alpha ,\beta \right)}}$$
where, the $\partial \left(\alpha ,\beta \right)$ function is the luminance intensity of a pixel at (*α*, *β*) coordinates and $\partial^{\prime }\left(\alpha ,\beta \right)$ is the luminosity enhanced image. At this point, the Adaptive Gamma Correction (AGC) technique is used to boost the brightness of a given image. The cumulative distribution function of the normalized histogram of the input fundus image is used to calculate the weight for the gamma correction method in this method. The brightness boost leads to a partial contrast improvement. However, in cases of low-contrast fundus images, a contrast enrichment technique is required to improve the image. Ben Graham’s method [47] is utilized then to finally improve the contrast of the retinograph images. Afterward, the J* plane of Jch color is combined, and then inverse transform is applied to reconstruct and visualize the enhanced image, as shown in Figure 3. This figure shows the results of the color contrast enhancement process. The results show a significant enhancement for the input fundus images, the first row in Figure 3, compared to the enhanced fundus images, the last row in Figure 3. This figure shows the visual example of light adjustment and contrast enhancement in JCh perceptual-oriented color space to enhance the retinograph images.

#### 3.2.2. Lesions Detection by Semantic and Instance Based Segmentation

Image segmentation is a critical and challenging aspect of image processing. Image segmentation divides an image into several regions with similar properties. Simply described, it is the process of separating a target from its surroundings in an image. Image segmentation algorithms are now progressing in a faster and more accurate direction. It is a very complicated task to segment HR-lesions from the retinograph image through simple image segmentation techniques. As a result, we have used the latest semantic- and instance-based segmentation techniques to detect HR-related lesions. These techniques are described in the upcoming sub-sections.

Semantic image segmentation, also known as pixel-level classification, is the task of grouping together image parts that belong to the same object class. Image segmentation is like pixel-level prediction in that it categorizes each pixel. For example, a glaucoma image has many regions with different colors, which are segmented pixel-wise. Recently, several deep learning studies [48] focus on semantic segmentation to do pixel-level classification. Convolution neural networks (CNN) also have excellent feature extraction capabilities; they do not need the manual extraction of image features or unnecessary image preprocessing. Currently, CNN has been used in medical image segmentation. Fully convolution layers neural network-based semantic segmentation is considered more successful in medial image segmentation tasks in recent times due to the advancement of deep learning technology. Semantic segmentation necessitates the extraction of dense features via a network with deep layers. However, the network with too many layers suffers from a vanishing gradient [49]. In our work, we use a shallow CNN model that consists only of convolution layers for feature extraction to alleviate this problem. The model we use is not computationally complex, and it works well on a limited dataset. The architecture of our proposed deep model includes three convolutional layers with a kernel size of (3 × 3) to get the feature map and a sigmoid function, as shown in Figure 4a. The proposed work uses images and masks (labeled images), as shown in Figure 4b,c. These images and their respective masks are used to train our model. The size of the mask used is the same as the input size of a retinograph image. These mask images are used as input into the deep learning model to extract features from the image mask and generate an output feature map. Afterward, a random forest (RF) machine learning classifier takes the output feature map from the deep learning model as an input to classify the pixels and obtain each pixel’s information. In general, classification tasks in the CNN model, such as VGG and ResNet fully connected layer, are used at the network’s end. The probability information about categories is obtained using the SoftMax layer. However, the category probability information is one-dimensional, and only the category of the entire image can be identified, not the variety of each pixel of an image. Therefore, a fully connected method is suitable for the image segmentation task [50].

Instance segmentation can be defined as the combination of object detection and semantic segmentation schemes. Following the object detection strategy, the instance segmentation achieves distinct class instances present in an image [51]. This shows the difference between instance segmentation compared to semantic segmentation. Several studies report instance segmentation. In our work, first, we generate the output mask using the U-Net model and then we apply a few image processing operators on processed images to get instance segmentation. The U-Net solves problems associated with general CNN networks used for medical image segmentation because it has a perfectly symmetric structure and skips connection. Medical images, unlike standard image segmentation, frequently contain noise and have blurred borders. Consequently, objects in medical imaging are too complicated to identify or distinguish, merely relying on low-level visual features. Meanwhile, it is also impossible to derive correct boundaries based on image semantic traits due to a lack of image detail information. On the other hand, the U-Net efficiently fuses low-level and high-level image characteristics by mixing low-resolution and high-resolution feature maps via skip connections, making it ideal for medical image segmentation tasks. The U-Net has become the industry standard for most medical image segmentation tasks, inspiring many significant advancements. The simple U-net architecture shown in Figure 4d is used to predict test images. This U-net model is famous end-to-end architecture composed of an encoder and decoder. The encoder part of the model is used to extract features from the input image of size (256 × 256). Also, the decoder portion is restored to the extracted feature to output the final segmented result having the same input size.

Instance segmentation can be defined as the combination of object detection and semantic segmentation schemes. Following the object detection strategy, the instance segmentation achieves the distinct class instances present in an image [51]. This shows the difference between instance segmentation and semantic segmentation. Several studies report instance segmentation. In our work, firstly, we generate the output mask using U-Net model, and then we apply a few image processing operators on processed images to get instance segmentation. The U-Net solves problems associated with general CNN networks used for medical image segmentation because it has a perfectly symmetric structure and skips connection. Medical images, unlike standard image segmentation, frequently contain noise and have blurred borders. Consequently, objects in medical imaging are complicated to identify or distinguish, merely relying on low-level visual features.

Meanwhile, it is also impossible to derive correct boundaries based on image semantic traits due to a lack of image detail information. On the other hand, the U-Net efficiently fuses low-level and high-level image characteristics by mixing low-resolution and high-resolution feature maps via skip connections, making it ideal for medical image segmentation tasks. The U-Net has become the industry standard for most medical image segmentation tasks, inspiring many significant advancements. The simple U-net architecture shown in Figure 4d is used to predict test images. This U-net model is famous end-to-end architecture composed of an encoder and decoder. The encoder part of the model is used to extract features from the input image of size (256 × 256). The decoder portion is restored to the extracted feature to output the final segmented result, having the same input size.

After generating the mask image using the U-net architecture, we have applied some image processing techniques (as shown in Figure 4e) to get the instance segmented image. First, we threshold the processed image using the Otsu algorithm. Then, we have used a morphological opening operation to remove noise from the image. Afterward, we apply the dilate function and distance transform function to determine the sure background and foreground based on the threshold value of 0.2. Then, we use the subtract operator to determine the unknown pixel in the processed image. Next, we define the markers to understand the connecting and not connecting pixel in our image. With the help of markers, the unknown pixel is also changed to a background pixel. Finally, the watershed algorithm is applied to segment each instance of an image. The image processing steps are then applied to enhance the detected boundaries. Finally in this paper, we have detected several distinguished features to classify five severity-levels of HR. In total, these distinct HR-related features are detected based on repeated experiments, and these features are statistically significant in the classification task.

#### 3.2.3. DenseNet Architecture for Classification

DenseNet169 is chosen as one of the best models in terms of accuracy and F1-score. Transfer learning is an efficient way [52,53] to achieve accurate results in classification problems using a small dataset. Those transfer learning (TL) algorithms are successfully applied to recognize the severity level of diabetic retinopathy (DR). Accordingly, we have been inspired by the past results based on the DenseNet169 model as a pretrain TL architecture to recognize five stages of HR. Hyper-tuning deep transfer learning models (DTL) can also boost performance. A DTL model based on DenseNet169 is proposed in this paper. A visual architecture of DenseNet is displayed in Figure 5. The proposed models used their learned weights on the ImageNet dataset, as well as a convolutional network’s structure, to extract features. Direct connections from all preceding layers to all subsequent layers are added to boost communication in the DenseNet169 model.

The feature concatenation can be mathematically explained as:(3)$$X_{l}=H_{l}\;\left(\left[x_{0},x_{1},x_{2},\ldots ,x_{l-1}\right]\right)$$

Here, $x_{0},x_{1},x_{2},\ldots ,x_{l-1}$ is a concatenation of a features map generated by a non-linear transformation Hl, which can be described as a composite function consisting of batch normalization (BN), supplemented by a rectified linear unit function (ReLU) and a convolution unit of (3 × 3). Dense blocks are formed in the network architecture for downsampling purposes, and they are separated by layers called transition layers, which consist of BN, a (1 × 1) convolution layer and, finally, a (2 × 2) average pooling layer. Because of its architecture, which considers feature maps as a global state of the network, DenseNet169 performs well even with a slower growth rate. As a result, each subsequent layer has access to all of the feature maps from the previous layers. Each layer adds k feature maps to the global state, with the total number of input feature maps at the first layer specified as:(4)$$f_{maps}\left(k_{l}\right)=k\;N\left(l-1\right)+k_{0}$$
where $k_{0}$ denotes the channels in the input layer. A bottleneck layer that is a (1 × 1) convolutional layer is added before each (3 × 3) convolution layer to increase computational efficiency by reducing the number of input feature maps, which are usually more than the output feature maps k. The bottleneck layer generates 4k feature maps. For classification, two dense layers with 128 and 64 neurons, respectively, are added. If the dimensions of function maps vary, DenseNet is divided into DenseBlocks, each with its own set of filters but the same dimensions. The transition layer uses downsampling to perform batch normalization; this is a crucial step in CNN.

Even though each DenseNet layer only produces *k* output feature maps, it usually has a lot more inputs. To reduce the number of inputs and thus boost the computational performance, a bottleneck (1 × 1) convolution can be added as a transition layer before each (3 × 3) convolution. A fundamental issue in DenseNet is the variance of the sizes of the feature maps. Consequently, it is impossible to group them and there is no difference if the grouping is concatenation or an addition. Therefore, DenseNets are divided into DenseBlocks, with the feature map dimensions remaining constant within each block but with the number of filters changing between them. From a conceptual perspective, the network is a series of parallel and serial calculations that map an input to an output. In this section, the way the proposed architecture imparts knowledge and learn is explained. This supervised learning allows the network to alter the way the steps are computed, allowing the output to be changed. Remembering that the general structure of supervised learning is achieved in the training phase through the training data such as understanding the output relative to each input allows you to determine the error about the network output.

A Denseblock deep-learning architecture is performed on various parts of the fundus image during each decoding step, and the HR-related features are determined by the previous hidden state and Denseblocks features. In the transition layer, the activation function creates convolution features that have pre-trained features, denoted as (X) with dimensions (16 × 16 × 1024). The DenseNet input consists of the current image plus the multiplication elementwise of X with a deterministic and soft visual-spatial feature outputting the subsequent DenseNet map each time. The DenseNet uses a piloted layer to combine the most salient portions of the network states. By using filters, the DenseNet architecture layers are very narrow, and the problem of gradient vanishes. To solve this issue, the DenseNet used each layer to have direct access to the gradients from the loss function and to the original input image. Therefore, we used the multiclass loss function in this paper.

The Kullback Leibler divergence loss (KLD-L) function in this paper before the SoftMax layer is used to backpropagate the DenseNet architecture. This KLD-L achieved better performance when compared to the categorical crossentropy loss function. Therefore, the KLD-L function provides how much information is lost as compared to other loss functions. In practice, the KLD-L measures how one probability distribution differs from a baseline distribution. Loss zero means that all distributions of features or data are identical. In a multi-class classification problem, this divergence loss function is a more common and preferable utilized function in the past, which is the same as the multi-class entropy function. This KLD-L function provided the loss at this epoch, which is obtained by calculating the cross-entropy loss. The network’s parameter is optimized by a backpropagation algorithm to minimize the loss of network output.

Figure 6a represents the DenseNet model loss vs. epochs plot. Finally, the SoftMax function is calculated by Equation (5). Let $H=\left(h_{1},\;h_{2},\;h_{3}\ldots h_{T}\right)$ be the $d_{h\times T}$ matrix of all the hidden states. The prediction mechanism (*Y*) outputs a (r × T) matrix of weights *W*, computed as follows:(5)$$Y=SoftMax\;(\tan h\;\left(W\cdot H\right))$$

## 4. Experimental Results

Experiments results are achieved on a statistical analysis of 1400 fundus images, including stage normal (NR) of 400, mild (MLD-HR) of 200, moderate (MOD-HR) of 200, severe (SEV-HR) of 200 and malignant (MLG-HR) of 400 by using sensitivity (SE), specificity (SP), accuracy (ACC) and region under the receiver operating curve (AUC) metrics. The set of 1400 hypertensive retinopathy images are obtained from six different online sources and one private medical hospital. By using data augmentation, we have doubled these 1400 HR images to 2800 HR images for better training and testing of the network model. In this paper, the five-stage based hypertensive retinopathy (HYPER-RETINO) system is developed to identify the severity level of hypertension. For better results and comparisons, all retinograph images are resized to (700 × 600) pixels. A computer Lenovo X1 carbon with 8 cores, 16GB RAM and 2GB Gigabyte NIVIDA GPU is used to develop the HYPER-RETINO program in Python. In our experimental tests, the losses for training, validation and accuracy over the validation set are measured. The parameter setting values used in the DenseNet architecture consist of (Optimizer: Adam, Learning Rate: 0.001, dropout rate: 0.2, Loss Function: Kullback Leibler divergence loss, Batch size: 10, Epochs: 80) to train the network. Figure 5 displays the loss and accuracy of the training and validation datasets when divided 50% and 50%, respectively, over the epochs. In addition, we have performed state-of-the-art comparisons with four recent HR systems. DL-based HR models are utilized, such as DenseHyper [31], Arsalan-HR [38], Kriplani-AVR [39] and Tag-Semantic [40] because of the ease of their implementation. Mostly, they were developed to detect lesions from retinograph images for diabetic retinopathy (DR) or HR stage classification.

Experimental results are analytically measured in terms of statistical metrics. The HYPER-RETINO system is evaluated and compared with other state-of-the-art hypertensive systems in terms of sensitivity (SE), specificity (SP), F1-score, Matthews correlation coefficient (MCC) and accuracy measures. In addition, to access the performance of the HYPER-RETINO system on five stages of hypertension, the comparisons are also performed by using different features, machine-learning and deep-learning (DL) models. An area under the receiver operating curve (AUC) is also plotted to show the performance. From Equation (6) to Equation (11), the sensitivity (SE), specificity (SP), F1-score, Matthews correlation coefficient (MCC) and accuracy are calculated.
(6)$$Senstivity\;\left(SE\right)=True\;Positive\;Rate\;\left(TPR\right)=\frac{TP}{TP+FN}$$
(7)$$Specificity\;\left(SP\right)=1-False\;Positive\;Rate\;\left(FPR\right)=\frac{FP}{FP+TN}$$
(8)$$F1-score=\frac{2TP}{2TP+FP+FN}$$
(9)$$Precision=\frac{TP}{TP+FP}$$
(10)$$Matthews\;correlation\;coefficient\;\left(MCC\right)\;=\frac{TP\times TN-FP\times FN}{\sqrt{\left(TP+FP\right)\left(TP+FN\right)\left(TN+FP\right)\left(TN+FN\right)}}$$
(11)$$Accuracy=\frac{TP+TN}{TP+FP+TN+FN}$$

The sensitivity (SE) statistical metric is also known as the true positive performance (TPR) measure, and it is calculated by Equation (6). The specificity (SP) is measured by the (1-FPR) metric and calculated by Equation (7). Whereas the precision metric in Equation (9) is used to find out the number of True Positive predictions. In Equation (9), the recall metric is measured to calculate the number of positive predictions divided by the number of positive class values. In Equations (8) and (10), the Matthews correlation coefficient (MCC) and F1 Score both convey the balance between the SE and SP. In all of these Equations (6)–(11), the true positive (TP) parameter decides the classifier recognized as a true positive case. False Positive is defined as when the actual class mark is negative (N) (FP). On the other hand, a true negative (TN) is counted if and only if both the predicted and actual class labels were N. When the classifier judgment is N but the actual mark is P, the false negative (FN) is counted. The value of the features set extracted is studied in this section, with the aim of demonstrating the relative contributions of different features to saliency analysis and HR-related lesion detection from retinograph images. A 10-fold cross-validation test is also used to compare the AUC to other state-of-the-art deep-learning architectures. In general, the AUC curve and confusion matrix were mostly used to access the index to assess the networks’ overall classification accuracy. The AUC value was found to range from 0.5 to 1.0. The system with the highest AUC outperforms the others. The HYPER-RETINO system’s efficiency has been statistically evaluated.

Firstly, an experiment is conducted to check the performance of the pre-processing step to enhance the contrast in a perceptual-oriented color space. The pre-processing method is implemented with the HYPER-RETINO system to obtain the higher classification and detection result of the fundus image. The preprocessing methods are obtained before inputting the retinograph images into the HYPER-RETINO to assess the precision and robustness of the HYPER-RETINO procedure. In fact, the pre-processing step results in the reduced overall complexity of the network model. The pre-processing methods of the retinograph images are composed of color space transform, light adjustment and enhanced contrast. After the pre-processing of the retinograph images, the distinct feature set is obtained by semantic and instance-based segmentation techniques. The detection of HR-related features is visually displayed in Figure 7. This figure clearly displays that the HR-related features are accurately detected. Without using pre-processing step and color space transform, the detection accuracy of the five stages of hypertensive retinopathy is significantly decreased as shown in Figure 8b,c.

The second experiment is performed after the image preprocessing to check the performance of the utilized U-Net model with a watershed transformation approach for the accurate pixel and HR lesion classification. It is concluded from this experiment that the proposed segmentation art results for both semantic and instance-based techniques are like the human expert ground truth masks using a small number of samples. Thus, the proposed semantic and instance segmentation framework is the best possible strategy for the retinal ophthalmologists to be implemented in the real-time clinical environment to identify and classify the HR lesions within the image.

The third experiment is conducted to check the performance of the HYPER-RETINO system based on a 10-fold cross-validation test. These results are depicted in Table 1 based on five stages of hypertensive retinopathy (HR). In Table 1, a total of 1400 images are utilized and measured in the five stages. On average, the SE of 90.5%, SP of 91.5%, ACC of 92.6%, PR of 91.7%, MCC of 61%, F1 of 92%, AUC of 0.92 and E of 0.60 are achieved. It shows that the proposed HYPER-RETINO system is outperformed to recognize five stages of HR such as MILD-HR, MODERATE-HR, Severe-HR, Malignant-HR and Normal (no sign of severity).

Fourth, the experiment is conducted to check the performance of the HYPER-RETINO system on the size of the retinograph image. During classification, though, it is difficult to enter the original retinograph image into the proposed HYPER-RETINO system, since the pixel size of the original retinograph image acquired in the dataset is of variable sizes. As a result, the image of this dataset is resized and cropped to (700 × 600) resolution. The effect of pixel sizes is mentioned in Table 2 based on the proposed HYPER-RETINO system on five stages of HR eye-related diseases. In practice, this table illustrates that the effect of pixel size decreases the accuracy of the system and increases computational efficiency. Compared to other sizes of the retinograph images, the size is fixed to (700 × 600) as a standard, because the network architecture of the HYPER-RETINO system has the highest accuracy in classifying the five stages of hypertension.

The fifth experiment is performed to check the performance of utilized pooling layers in the deep learning architecture based on different sizes based on average and maximum pooling layers. An average pooling technique is used to reduce the error caused by the increase in the variance of the estimated value in the limited size of the neighborhood, whereas the maximum pooling technique is used to select maximum features in the neighborhood. Therefore, Table 3 is derived based on average and maximum pooling techniques. To perform these comparisons, the HYPER-RETINO architecture is used to recognize five stages of the hypertension in terms of 6-pooling, 7-pooling, 8-pooling, 9-pooling and 10-pooling average and maximum layers. Compared to the average pooling technique, the maximum pooling is provided to get high performance (SE of 91.5%, SP of 89.5%, ACC of 90%, PR of 89.3%, MCC of 62%, F1 of 88%, AUC of 0.89 and E of 0.64). Significantly, the increase in the pooling layer provides lower performance. As a result, the six-pooling layers with the maximum strategy are used to implement the HYPER-RETINO system.

The sixth experiment is performed to check the performance of the HYPER-RETINO architecture over different sizes of dense blocks. It became evident by experiments that the dense connection provides better results compared to other deep network models with the same depth. This sixth experiment is very important to understanding the property of model overfitting. In practice, the denseblock is used to reduce overfitting and solve the gradient vanishing problem without using optimization. Compared to other sizes of blocks, the 4-denseblock outperforms on the identification of five stages of HR and achieved SE of 90.5%, SP of 91.5%, ACC of 92.6%, PR of 91.7%, MCC of 61%, F1 of 92%, AUC of 0.92 and E of 0.60. Therefore, the increased size of denseblock provides lower performance results. The accuracy of the HYPER-RETINO network for the five stages of the classification of hypertension is not improved as the number of denseblock increases.

The seventh experiment is conducted to compare the proposed system with other state-of-the-art systems based on two stages and five stages of the severity level of hypertensive retinopathy (HR). Table 4 and Table 5 are used to describe those comparisons on the selected dataset of two-stage and five-stage HR, and corresponding graphs are visually displayed in Figure 9. In comparison to other state-of-the-art approaches for HR recognition that use deep learning (DL) models, few research efforts use DL approaches for classifying HR from fundus videos. These four DL-based HR models are utilized: DenseHyper [31], Arsalan-HR [38], Kriplani-AVR [39] and Tag-Semantic [40]. This is because of their ease of implementation. Compared to other systems, the proposed system outperformed in both two stages and five stages-based recognition of HR when the eye is a diagnosis by retinograph images. The results are explained in the subsequent paragraphs.

Table 5 shows the performance of a proposed HYPER-RETINO method compared to other methods. As can be observed from this table, the Arsalan-HR [38] model obtained a SE of 78.5%, SP of 81.5%, ACC of 80% and AUC of 0.80. Furthermore, the Arsalan-HR approach is based on a simple CNN model that is unable to derive useful and simplified features to distinguish HR and non-HR. The identification accuracy of 98.6 percent is reported in Arsalan-HR [38] due to the use of small training and the testing dataset of HR. The authors in the Kriplani-AVR [39] scheme used the aggregate residual neural network (ResNeXt) model as a classifier, and they achieved 94% of accuracy on the structured analysis of the retina (STARE) dataset. For segmentation and feature extraction, the device employs image processing techniques. The classifier’s input is a function vector made up of the A/V Ratio rather than the fundus picture. To equate the proposed HYPER-RETINO method to Kriplani-AVR [39], we follow the same steps and tested it on the model Kriplani-AVR. On average, the Kriplani-AVR model achieves SE of 74.5%, SP of 73.5%, ACC of 74% and AUC of 0.74, which is slightly higher than the Arsalan-HR [38] system.

Recently, the authors in Tag-Semantic [40] have developed a deep model based on pretrained CNN with an interpretable Tag-Semantic approach to recognize the stage of diabetic retinopathy (DR) instead of hypertensive retinopathy (HR). However, this Tag-Semantic system can be used to detect the two stages of HR in case of the detection of the HR-related lesions. Therefore, this approach is included in the comparisons, as this system is closed to the proposed HYPER-RETINO system. The Tag-Semantic approach was used to extract lesions from retinograph images by using CNN and 3 layers of CNN models and then using the guided backpropagation approach to adjust weights in the network. Compared to Arsalan-HR [38] and Kriplani-AVR [39], the Tag-Semantic system outperforms and achieves SE of 80.5%, SP of 79.5%, ACC of 81% and AUC of 0.82 values. Moreover, in the DenseHyper [31] system, the authors obtained SE of 81.5%, SP of 82.5%, ACC of 83% and AUC of 0.84, which is greater compared to previous methods. In comparison, the SE, SP, ACC and AUC of our proposed HYPER-RETINO model were all higher, with SE of 93%, SP of 90.5%, ACC of 92.5% and AUC of 0.92. As a result, they have a high level of precision. On the other hand, the HYPER-RETINO system developed in this paper has been evaluated with a large dataset. Thus, the HYPER-RETINO achieves an accuracy of classification of 92.5%.

A visual example in Figure 7 shows the detection and recognition of the five stages of HR based on the proposed semantic and instance segmentation techniques in a DenseNet DL architecture. This figure shows the various stages of HR in case of a normal (no sign of abnormality), mild HR, moderate HR, severe HR, and malignant HR. Next, the eighth experiment is conducted to evaluate the performance of the proposed Hyper-Retino system on a large dataset. To perform this experiment, we have applied the data augmentation technique to increase the size of the dataset and gold standard masks from 1400 to 2800 retinograph images. However as mentioned before, we have performed the data augmentation technique in an offline fashion. In the data augmentation step, the flipping (horizontally + vertically) and rotation from 90 degrees to 180 degrees without scaling are applied. This increases the number of training images from 1400 to 2800. As shown in Figure 8b,c, the Hyper-Retino achieved a detection accuracy of HR for five stages up-to-the-mark with and without the preprocessing step. In addition, the visual representation of state-of-the-art comparisons of the HYPER-RETINO system in terms of the AUC curve with 2800 HR images is displayed in Figure 9 with a preprocessing step. It noticed that the preprocessing step contributes toward getting high classification accuracy compared to no preprocessing step as shown in Figure 10. In addition, a visual representation of state-of-the-art comparisons indicates that the proposed HYPER-RETINO system is better than other detection systems.

## 5. Discussion

A few computerized systems were developed in the past to recognize hypertensive retinopathy (HR) by using two stages such as HR and non-HR. Those systems, on the other hand, focused on extracting HR-related features by using handcrafted techniques and deep-learning (DL) models. Furthermore, defining advanced features to understand multistage HR (five-stage) is challenging for DL models. The classification accuracy is greatly increased when deep-feature methods are used. In this paper, a new hypertensive retinopathy (HYPER-RETINO) mechanism has been formed to grade the HR into five stages such as regular (NR), mild (MLD-HR), severe (MOD-HR), severe HR (SEV-HR) and malignant (MLG-HR). To create the HYPER-RETINO grading system, four major phases are used, such as a pre-processing step to improve HR-related lesions in perceptual-oriented color spaces (as shown in Figure 10), an initial semantic-based segmentation of HR-related lesions, the refinement of instance-based segmented regions and the classification of lesions by a pretrained DenseNet architecture.

In this paper, the derived HR-related features set is effective in recognizing the five stages of hypertensive retinopathy (HR). The HYPER-RETINO system is developed by motivating ophthalmologists to quantify the different types of HR-related lesions in retinal images. The accurate detection of retinal lesions is a challenging problem due to variations across patients, image intensity inhomogeneity, irregular shape and appearance of lesions. As a result, the DenseNet architecture on distinct extracted regions is applied to recognize the severity of HR.

According to a comprehensive literature review of retinopathy hypertension (HR), a single and accurate approach to recognize the five stages of the severity level of this disease is not available. Moreover, the automatic identification of multiple HR-related lesions at the pixel level has not been utilized before. Based on the knowledge, this is the first work of research based on a deep-learning (DL) architecture that can detect hemorrhages, microaneurysms, exudates, etc. automatically from retinal fundus images. Overall, the HYPER-RETINO approach outperforms the most current alternative approaches according to the experimental findings, which are based on six freely available datasets along with one private dataset. The proposed approach is capable not only of detecting lesions in images, but it can also be reliable in locating and measuring the size of those lesions to understand the severity level of this disease. It is worth noting that the statistical assessment metrics show that the HYPER-RETINO system outperforms deep learning-based methods, which have recently attracted a lot of attention.

With no complex parameter tuning or training data collection, the semantic and instance-based segmentation technique can detect both dark and light lesions to classify five stages of HR. The object size (which distinguishes the MA from the HE) or strength magnitude may be used to differentiate these lesions (which discriminate between the dark and bright lesions). The proposed approach can detect the vasculature, optic disk, macular, retinal vessel segmentation, optic disk recognition, macular extraction and anomalies. The proposed approach (HYPER-RETINO) would make a major contribution to health informatics by providing an effective instrument for retinal image analysis.

As stated in the related work Section 2, a few classification systems were developed in the past for the classification of two-stage (HR and non-HR) related eye disease. Those approaches are developed based on the latest trend, namely, deep-learning methods, rather than conventional machine learning approaches. When HR automated systems are designed using conventional approaches, there are several major difficulties. The first difficulty is that identifying and extracting relevant HR-related lesion features from retinograph images to determine HR properties using sophisticated pre- or post-image processing methods is extremely difficult. There are no datasets available with clinical expert labeling to identify such HR-related lesion patterns to train and validate the network. As a result, computerized programs have a hard time detecting disease features. The authors use manual hand-crafted features to train the network and compare the output of conventional and new deep-learning models, according to the literature. Consequently, finding the best features necessitates the use of an automatic strategy. Deep-learning algorithms have the highest outcomes as compared to conventional approaches. Many different models, on the other hand, use qualified models created from scratch to learn features automatically, but they all use the same weighted scheme at each step. Layers can find it difficult to move weights to higher levels of the network for specific decisions.

As seen in Table 4 and Table 5, the HYPER-RETINO method outperformed the Triwijoyo-CNN-2017 [7] and Pradipto-CNN-RBM-2017 [33] state-of-the-art HR recognition systems in terms of precision as compared to general DL models. This is because the HYPER-RETINO framework was built using the DenseNet architecture with a pretrained strategy approach and is based on qualified features. In addition, the DenseNet architecture was pretrained to extract, localize and specialize features. Moreover, the semantic- and instance-based segmentation provided the best results in the proposed HYPER-RETINO system to detect various HR-related lesions.

The HYPER-RETINO method for HR identification can be enhanced in the future by offering a larger dataset of retinograph images obtained from various sources. To improve the model’s classification accuracy, it may be possible to use hand-crafted features instead of only deep features. Since then, several researchers have used the saliency maps technique to segment DR-related lesions and then used a trained classifier to separate those lesions from retinograph images. Only the segmentation phase has been done in those studies. These saliency maps will be combined in the future to improve HR eye-related disease recognition accuracy. Furthermore, the HR severity varies. However, the extraction of those HR-related lesions with different thresholds will be used to detect the disease level of HR. As a result, it can be a useful option for clinicians to employ them to solve the problem of hypertension. Two expert ophthalmologists helped us in evaluating the performance of the proposed automatic classification system. Based on the system’s recommendation, they informed us that the obtained results of the automatic analysis of hypertensive retinopathy are useful for them in the diagnosis of the diseases and help them significantly.

## 6. Conclusions

Only a few computerized systems have been built in the past to identify two-stage diabetic hypertensive retinopathy (HR) based on retinography and deep-learning architectures. However, these systems are dependent on image processing to extract characteristics from several HR-related lesions (ratio of arteriolar to venular diameter, blood vessels, optic disk, cotton wool spot, microaneurysms, tortuosity and hemorrhages) and to classify them by machine-learning algorithms. As a practice, the recognition mechanism for hypertension is made up of domain-expert knowledge of feature selection and image processing. According to our knowledge, there are few frameworks presented in the past for the two stages (HR versus non-HR) based recognition based on deep learning (DL) models. These systems have been tested on small datasets without preprocessing steps. Therefore, it is hard to use them as a screening method for the identification of the severity level of HR. Moreover, the precision of classification is not up-to-the-mark. In this paper, a novel computerized hypertensive retinopathy (HYPER-RETINO) system is developed to detect five stages of HR based on semantic- and instance-based dense layer architecture and a pre-training strategy to solve the above-mentioned problems. At last, the HYPER-RETINO system can detect HR and it can help to assist the ophthalmologist as well as facilitate mass screening. The applicability of the HYPER-RETINO method to reliably diagnose hypertensive retinopathy is verified by experimental findings. In future works, we will try to develop a differentiation system between diabetic retinopathy (DR) and hypertensive retinopathy (HR).

## Figures and Tables

**Figure 1 sensors-21-06936-f001:**
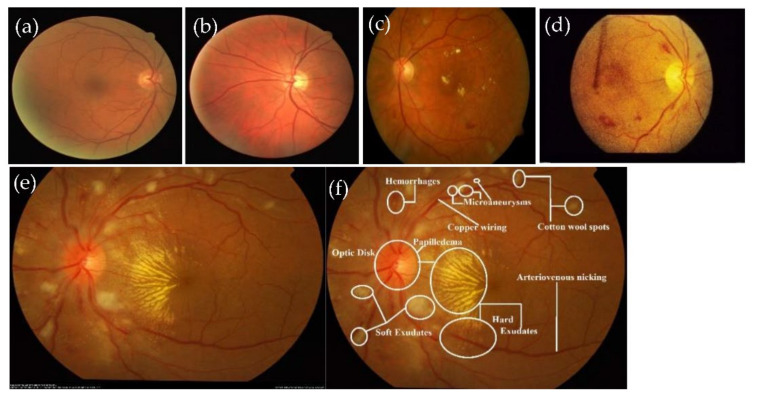
A visual example of different stages of HR, where figure (**a**) shows the normal (no sign of abnormality), (**b**) mild, (**c**) moderate, (**d**) severe and (**e**) malignant hypertensive retinopathy fundus, including figure (**f**), which shows the HR-related lesions. The grades of HR with (**a**) Grade 0: normal HR, (**b**) Grade 1: MID-HR with arteriolar narrowing (white arrow), copper wiring (black star) and AV nicking (black arrow). (**c**) Grade 2: MOD-HR with features of MID-HR+ cotton wool spots hemorrhages. (**d**) Grade 3: SEV-HR with features of MOD-HR and optic disc swelling, and (**e**) Grade 4: MLG-HR with features SEV-HR+ Papilledema and (**f**) normal retinograph.

**Figure 2 sensors-21-06936-f002:**
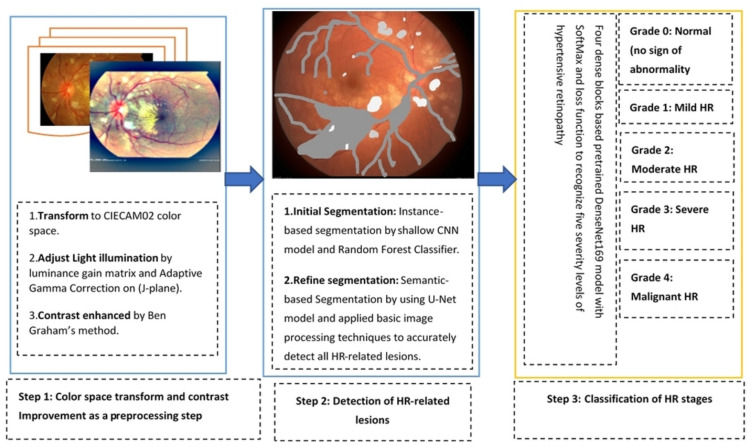
A systematic flow diagram of the HYPER-RETINO system for the diagnosis of five-stage of HR- eye related disease classification.

**Figure 3 sensors-21-06936-f003:**
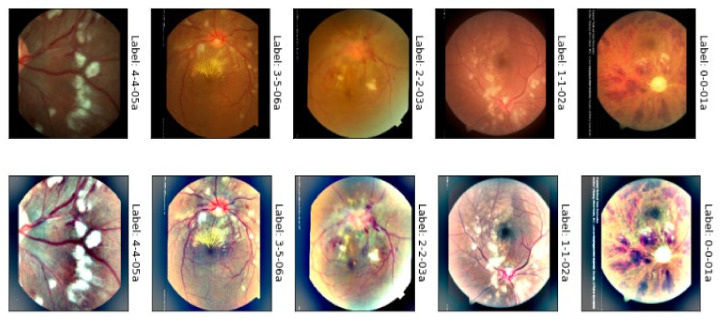
A visual example of the proposed light adjustment and contrast enhancement in the CIECAM02 (JCh) perceptual-oriented color space of malignant hypertensive retinopathy.

**Figure 4 sensors-21-06936-f004:**
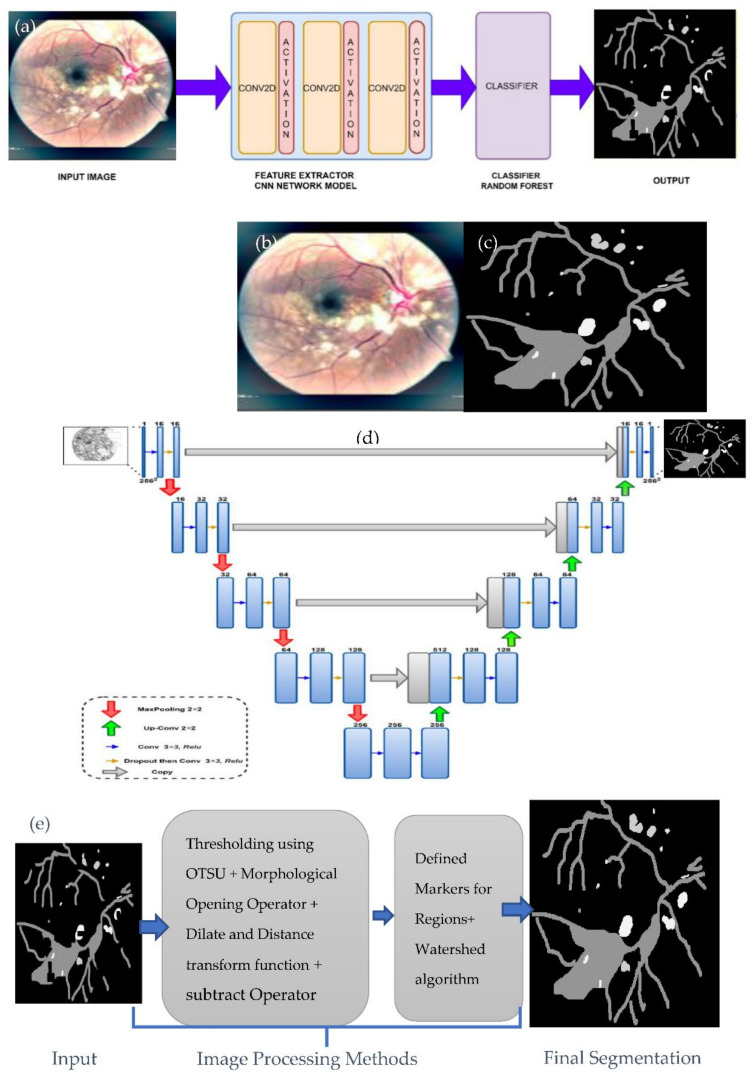
A visual example of HR-related lesions segmentation steps. Where figure (**a**) shows the shallow CNN architecture with the Random Forest (RF) classifier used for sematic-based segmentation, figure (**b**) represents the malignant-HR image, figure (**c**) shows the corresponding mask, figure (**d**) shows the design architecture used for semantic-based segmentation through the U-Net model to refine detection results and, finally, figure (**e**) indicates the image processing steps to get the final regions of HR-related lesions.

**Figure 5 sensors-21-06936-f005:**
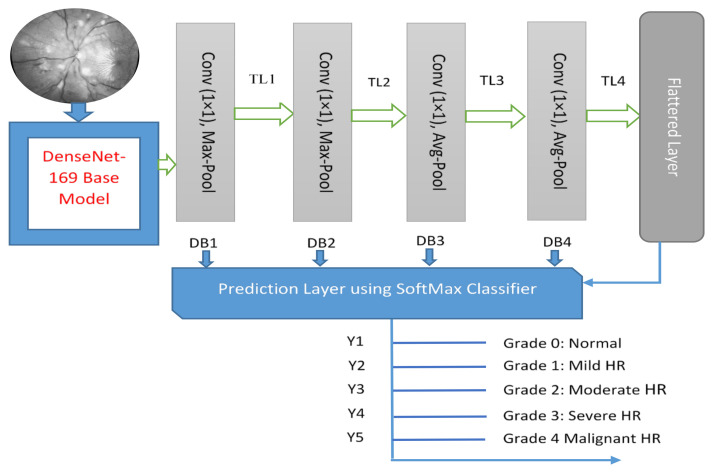
A visual architecture of proposed HYPER-RETINO transfer learning developed to predict five stages of diabetic HR when the diagnosis is through retinograph images, where TL shows the transition layer and DB represents the dense block.

**Figure 6 sensors-21-06936-f006:**
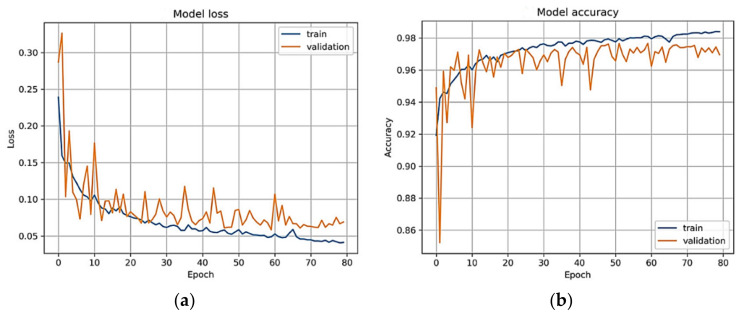
Training and validation dataset used to calculate the loss and accuracy of the final DenseNet model. The model received the retinograph as an input and is trained until there is convergence on the augmented dataset. (**a**) Model loss; (**b**) model accuracy.

**Figure 7 sensors-21-06936-f007:**
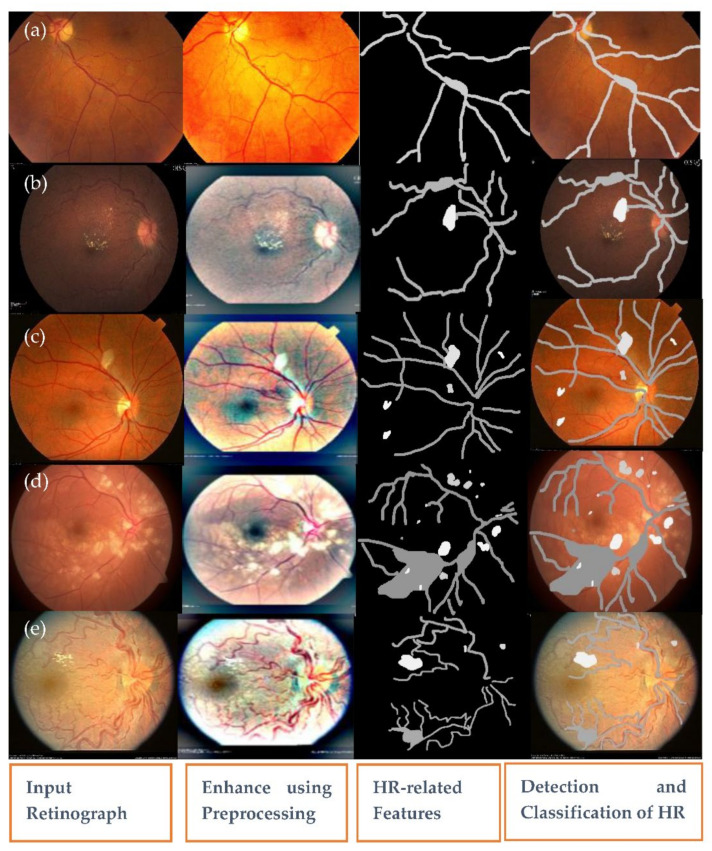
A visual example of the proposed semantic and instance segmentation of HR lesions where there is (**a**) no sign of abnormality, (**b**) mild HR, (**c**) moderate HR, (**d**) severe HR and (**e**) malignant HR.

**Figure 8 sensors-21-06936-f008:**
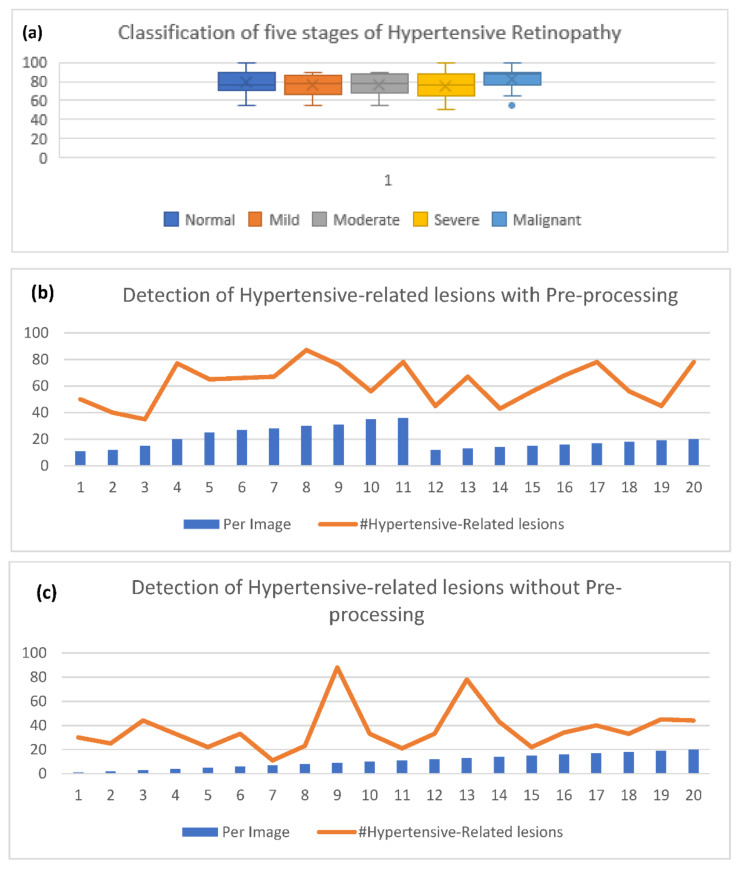
Experimental results obtained with pre-processing and without pre-processing of the proposed model to predict and detect five stages of hypertensive retinopathy, where (**a**) shows the five stages of the severity level detection result, (**b**) presents the accuracy of the detected HR-related lesions related to each image with preprocessing and (**c**) without preprocessing on 2800 retinograph images by data augmentation.

**Figure 9 sensors-21-06936-f009:**
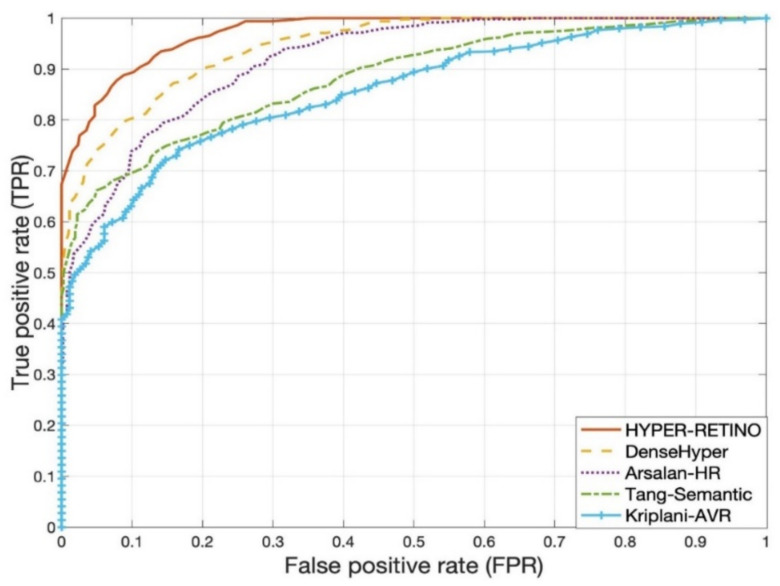
Visual representation of State-of-the-art comparisons of the HYPER-RETINO system in terms of the AUC curve five stages of HR from 2800 HR images with a preprocessing step.

**Figure 10 sensors-21-06936-f010:**
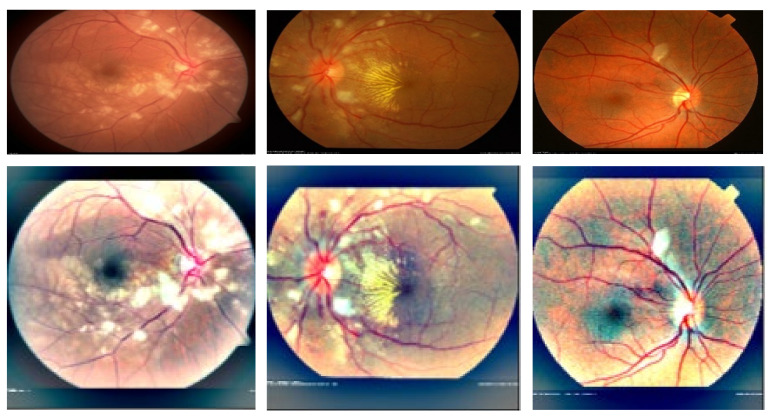
A visual example of the preprocessing step to enhance retinograph images related to malignant hypertensive retinopathy.

**Table 1 sensors-21-06936-t001:** Results of the HYPER-RETINO system on 1400 retinograph images for five stages of HR.

No.	Data Set	SE%	SP%	ACC%	PR%	MCC%	F1%	AUC	E
1	Mild	90.4	91	91	90.4	60	88	0.89	0.56
2	Moderate	88.2	88.5	87.5	88.2	58	89	0.92	0.60
3	Severe	87	89.4	89.5	89.5	61	91	0.93	0.62
4	Malignant	89.7	90.8	95	89.7	65	93	0.90	0.57
5	Normal	94	93	91	94	60	95	0.91	0.58
Average		90.5	91.5	92.6	91.7	61	92	0.92	0.60

MCC: Matthews correlation coefficient, SE: Sensitivity, SP: Specificity, F1: F1 score, ACC: Accuracy, E: Training errors and AUC: Area under the receiver operating curve.

**Table 2 sensors-21-06936-t002:** Results obtained by varying the image input size and masks of the HYPER-RETINO system on 1400 retinal images.

No.	Data set	SE%	SP%	ACC%	PR%	MCC%	F1%	AUC	E
1	600 × 700	93	90.5	91.5	92.6	60	90	0.92	0.60
2	512 × 512	92.5	89	88.5	89.2	59	87	0.89	0.65
3	500 × 500	91	88	87.5	86.4	58	84	0.83	0.75
4	450 × 450	89	86	86.5	85.2	56	82	0.82	0.80
5	400 × 400	85	83	83.5	80.6	54	79	0.78	0.90
6	380 × 380	75	71.5	72.5	73.6	51	71	0.70	0.905

MCC: Matthews correlation coefficient, SE: Sensitivity, SP: Specificity, F1: F1 score, ACC: Accuracy, E: Training errors and AUC: Area under the receiver operating curve.

**Table 3 sensors-21-06936-t003:** Results achieved by varying input pooling size of the HYPER-RETINO system.

No.	Architecture	SE%	SP%	ACC%	PR%	MCC%	F1%	AUC	E
1	6-P-ND-M	91.5	89.5	90	89.3	61	88	0.89	0.64
2	6-P-D-A	89.2	88.1	88	89.2	58	87	0.88	0.67
3	7-P-ND-M	87.3	86.5	86	87.3	57	86	0.87	0.69
4	7-P-D-A	86.4	85.2	85.4	86.4	55	85	0.86	0.72
5	8-P-ND-M	84.1	83.5	83.1	84.1	54	84	0.84	0.74
6	8-P-D-A	80.6	79.1	79.3	80.6	53	81	0.80	0.76
7	9-P-ND-M	78.7	76.5	77.5	78.7	51	79	0.78	0.78
8	9-P-D-A	75.5	74.6	74.4	75.5	48	75	0.75	0.80
9	10-P-ND-M	73.3	72.5	72.1	73.3	46	72	0.73	0.83
10	10-P-D-A	72.2	71.1	71.6	72.2	45	71	0.72	0.85

-P-ND-M: Pooling-no dropout-maximum layers, -P-D-A: Pooling-dropout-Average layers, MCC: Matthews correlation coefficient, SE: Sensitivity, SP: Specificity, F1: F1 score, ACC: Accuracy, E: Training errors and AUC: Area under the receiver operating curve.

**Table 4 sensors-21-06936-t004:** State-of-the-art comparison analysis based on 1400 HR images to recognize two stages in the pattern form SE%/SP%/ACC%/AUC.

Methods	NR	MLD-HR	MOD-HR	SEV-HR	MLG-HR
Arsalan-HR [38]	84.5/83.5/83.5/0.84	85.5/85.5/85.5/0.85	84.5/85.5/84.5/0.85	80.5/81.5/82.5/0.80	84.5/83.5/83.5/0.84
Kriplani-AVR [39]	82.5/81.5/82.5/0.82	83.5/82.5/83.5/0.83	83.5/82.5/82.5/0.82	80.5/81.0/82.0/0.81	82.5/81.5/82.5/0.82
Tag-Semantic [40]	83.1/82.2/81.5/0.83	84.1/85.2/84.5/0.84	83.1/82.2/81.5/0.82	82.1/82.2/81.5/0.82	83.1/82.2/81.5/0.83
DenseHyper [31]	85.5/83.5/83/0.84	87.5/84.5/83/0.86	85.6/84.5/84/0.85	82.5/84.5/84/0.82	84.5/83.5/83/0.84
HYPER-RETINO	94.1/93.4/91/0.92	90.0/91.4/91/0.89	88.5/88.2/87.5/0.92	90.5/93.5/96/0.93	89.5/90.5/95/0.90

NR: Normal retinograph images, MLDR: Mild hypertensive retinopathy, MOD-HR: Moderate hypertensive retinopathy, severe HR (SEV-HR) and MLG-HR: Malignant hypertensive retinopathy.

**Table 5 sensors-21-06936-t005:** State-of-the-art comparison analysis based on 1400 HR images to recognize five stages of HR.

No.	Methods	SE	SP	ACC	AUC
1	Arsalan-HR [38]	78.5%	81.5%	80%	0.80
2	Kriplani-AVR [39]	74.5%	73.5%	74%	0.74
3	Tang-Semantic [40]	80.5%	79.5%	81%	0.82
4	DenseHyper [31]	81.5%	82.5%	83%	0.84
5	HYPER-RETINO	93%	90.5%	92.5%	0.92

SE: Sensitivity, SP: Specificity, ACC: Accuracy, AUC: Area under the receiver operating curve.

## Data Availability

Not applicable.

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
