# Peer review of "An Automatic Detection and Classification System of Five Stages for Hypertensive Retinopathy Using Semantic and Instance Segmentation in DenseNet Architecture"

_sensors, 2021, doi:10.3390/s21206936_

Round 1

Reviewer 1 Report

This paper introduces a novel pipeline for the detection and classification of the grade of the Hypertensive Retinopathy (HR) pathology.

Even though the paper is interesting and the achieved results are promising, the authors should address and solve the following concerns before a possible publication.

  • In the Abstract the authors stated that “On average, a 10-fold cross-validation test obtained sensitivity of 24 90.5%, specificity of 91.5%, Accuracy of 92.6%, Precision of 91.7%, Recall of 90%, F1-score of 92% 25 and area-under-the-curve of 0.915 on 1400 HR images.”. However,  sensitivity and recall is the same metric. Please, double-check the calculated metric.
  • The Related work Section should be improved by citing/discussing other state-of-the-art Deep Learning architecture (some possible references are [Isola et al. "Image-to-image translation with conditional adversarial networks", 2016], [Ronneberger et al. “U-net: Convolutional networks for biomedical image segmentation”, 2015], [Milletari et al. "V-Net: fully convolutional neural networks for volumetric medical image segmentation", 2016], [Badrinarayanan et al. "SEGNET: a deep convolutional encoder-decoder architecture for image segmentation", 2017], [Rundo et al. "CNN-based prostate zonal segmentation on T2-weighted MR images: a cross-dataset study", 2020].
  • Please, double-check the mathematical formulation throughout the manuscript (matrices and vectors must be in bold, etc).
  • Please, remove vague sentences such as “They achieve a reasonable accuracy”, “Several deep learning studies [45] focus on semantic segmentation”, and  “Therefore, in recent times CNN has been used in medical image segmentation”. When does the accuracy can be considered reasonable?
  • English should be revised in any section of the paper because some typos are present throughout the manuscript and the used English should be polished by further proofreadings (e.g., Kullback Leibler Divergence Los (KLD-L) -> Kullback Leibler Divergence Loss (KLD-L) and then KLD-L should be used instead of KLD-L loss. “The subscript for the A few computerized systems that recognize HR by using two stages have been developed (HR versus non-HR)”).
  • Please, introduce the acronyms just once in the Abstract and once in the main text. In addition, other acronyms were not introduced (DCCT, ETDRS). The acronyms must be explained the first time they are used. 
  • Please, double-check the figures. For instance, the caption of Figure 7 is “A visual example of proposed semantic and instance segmentation of HR lesions where (a) normal no sign of abnormality, (b) mild HR (c) moderate HR (d) severe HR, and (d) malignant HR”. However, panels (a)-(d) are not indicated in the picture.

Author Response

Original Manuscript ID:  ID: Sensors-1409049              

Original Article Title: ‘An Automatic Detection and Classification System of Five Stages for Hypertensive Retinopathy using Semantic and Instance Segmentation in DeepNet Architecture'

To: Editor in Chief,

Sensors, MDPI

Re: Response to reviewers

Dear Editor,

Many thanks for insightful comments and suggestions of the referees. Thank you for allowing a resubmission of our manuscript, with an opportunity to address the reviewers’ comments.

We are uploading (a) our point-by-point response to the comments (below) (response to reviewers), (b) an updated manuscript with yellow highlighting indicating changes, and (c) a clean updated manuscript without highlights (PDF main document).

By following reviewers’ comments, we made substantial modifications in our paper to improve its clarity and readability. In our revised paper, we represent the improved manuscript.

We have made the following modifications as desired by the reviewers: Please see the attachment.

Best regards,

Corresponding Author,

Dr. Qaisar Abbas (On behalf of other author),

Associate Professor.

Reviewer 2 Report

The manuscript proposes a system named HYPER-RETINO to automatic diagnose the stages of Hypertensive Retinopathy (HR). A preprocessing step is integrated to adjust light illumination and enhance the contrast in a perceptual-oriented color space. A novel semantic and instance-based segmentation mechanism is introduced to classify and identify the HR-related lesion’s pixels and regions. Experiment results on 1400 fundus images and comparison with other related works show the advantage of HYPER-RETINO.

Major comments:

  1. The Y-axis in Figure 8 (b) and (c) are different, considering that they are for comparison, please keep them consistent to avoid confusions.
  2. What’s the reason to choose different percentages for training / testing data sets? The authors used 40%/60%, 50%/50%, 60%/40%, 30%/70% for different tests.
  3. The authors compared the proposed methods with reference 31, 51-53. But reference 51-53 were not mentioned in “Chapter 2. Literature Review”.
  4. It’s not clearly explained what the rectangles are in the output of Figure 4(a) and input of Figure 4(e). If they are masks, how to determine their sizes?

Minor comments:

  1. Please double check all the figures and tables:

Some figures are not well aligned: Figure 1, Figure 2, Figure 5, Figure 7, Figure 10

In Figure 8 (a), the maximum value of Y-axis should be 100

Some tables are too wide to fit in a single page, the authors should consider how to re-arrange them

In Figure 4(e), the words are not easy to read

  1. Please double check the grammar, typos and unclear expressions, e.g.,

Figure 2: spce -> space

Line 403-406: The proposed model is famous end-to-end architecture composed of an encoder and decoder. The encoder part of the model extract feature from the input image of size 256*256, and the decoder portion restored the extracted feature to output the final segmented result having the same input size. -> grammar error

The caption of Figure 7: two (d) exist. And it’s better to add the labels on the image.

Line 694-695: In practice, this table illustrate that the effect of pixels size decreases the accuracy of the system and increases computational efficiency. -> grammar error

Line 922-923: In this paper, the derived HR-related features set is affective to recognize five-stages of hypertensive retinopathy. -> effective

Line 951-952: A visual example of preprocess retinograph images for malignant hypertensive retinopathy -> preprocessing

Line 1013-1014: Two expert ophthalmologists helped us in evaluate the performance of proposed automatic classification system -> evaluating

Author Response

(The authors gave the same response as above.)

Round 2

Reviewer 2 Report

The manuscript proposes a system named HYPER-RETINO to automatic diagnose the stages of Hypertensive Retinopathy (HR). A preprocessing step is integrated to adjust light illumination and enhance the contrast in a perceptual-oriented color space. A novel semantic and instance-based segmentation mechanism is introduced to classify and identify the HR-related lesion’s pixels and regions. Experiment results on 1400 fundus images and comparison with other related works show the advantage of HYPER-RETINO.

Minor comments:

  1. Some figures are still not well aligned: Figure 1, 10
  2. Some boundaries of text boxes in Figure 5 are not erased.
  3. The line labels are overlapped with all the tables in the manuscript, please double check.

Author Response

Reviewer 1:

Comment - (1) Some figures are still not well aligned: Figure 1, 10.

Response 1: Yes, you are right. We have redrawn the figure as suggested by you in the modified version of the paper. Now, the Figures 1 and 10 in the revised paper does not have any alignment issues.

Thank you for nice to improve the quality of our paper.

Comment - (2) •Some boundaries of text boxes in Figure 5 are not erased.

Response 2: As suggested by reviewer #1, we have changed figure 5 to remove this problem. In the updated version of the paper, we have erased the boundaries of some text boxes. Thank you for this comment.

Comment - (3) • The line labels are overlapped with all the tables in the manuscript, please double-check.

Response 3: Yes, you are right. In the revised paper, we have to set about all the tables. Now, in pdf and word formats, there is no overlapping to line numbers to the table text.

Thank you once again.